# Potential Benefits of Omega-3 Polyunsaturated Fatty Acids (N3PUFAs) on Cardiovascular Health Associated with COVID-19: An Update for 2023

**DOI:** 10.3390/metabo13050630

**Published:** 2023-05-05

**Authors:** Louise Weiwei Lu, Siew-Young Quek, Shi-Ping Lu, Jie-Hua Chen

**Affiliations:** 1School of Biological Sciences, University of Auckland, Auckland 1010, New Zealand; 2Food Science, School of Chemical Sciences, The University of Auckland, Auckland 1010, New Zealand; sy.quek@auckland.ac.nz; 3Riddet Institute, New Zealand Centre of Research Excellence for Food Research, Palmerston North 4474, New Zealand; 4Pharma New Zealand PNZ Limited, Hamilton 3210, New Zealand; will@pharma-newzealand.com; 5Institute for Innovative Development of Food Industry, Shenzhen University, Shenzhen 518060, China; jiehua.chen@szu.edu.cn; 6Shenzhen Key Laboratory of Marine Microbiome Engineering, Institute for Advanced Study, Shenzhen University, Shenzhen 518060, China

**Keywords:** N3PUFA, EPA, DHA, omega-3, cardiovascular, coagulation, thrombosis, COVID-19, inflammation, oxidative stress

## Abstract

The accumulating literature demonstrates that omega-3 polyunsaturated fatty acid (n-3 polyunsaturated fatty acid, N3PUFA) can be incorporated into the phospholipid bilayer of cell membranes in the human body to positively affect the cardiovascular system, including improving epithelial function, decreasing coagulopathy, and attenuating uncontrolled inflammatory responses and oxidative stress. Moreover, it has been proven that the N3PUFAs, eicosapentaenoic acid (EPA) and docosahexaenoic acid (DHA), are precursors of some potent endogenous bioactive lipid mediators that mediate some favorable effects attributed to their parent substances. A dose–response relationship between increased EPA and DHA intake and reduced thrombotic outcomes has been reported. The excellent safety profile of dietary N3PUFAs makes them a prospective adjuvant treatment for people exposed to a higher risk of cardiovascular problems associated with COVID-19. This review presented the potential mechanisms that might contribute to the beneficial effects of N3PUFA and the optimal form and dose applied.

## 1. Introduction

Coronavirus disease 2019 (COVID-19), which is caused by severe acute respiratory syndrome coronavirus 2 (SARS-CoV-2), has been linked to 6.7 million deaths and 657.9 million reported cases of infection by January 2023 [1]. Mounting evidence of endothelial dysfunction in the autopsies of COVID-19 patients supports the hypothesis that there is an association between endothelial dysfunction (e.g., systemic hypertension, cardiovascular diseases, diabetes, and obesity) and the SARS-CoV-2 infection [2], as well as the post-manifestations of COVID-19 (i.e., long COVID-19) [3]. The endothelium is a single-cell sentinel layer that lines the innermost blood vessels with many functions, such as acting as a mechanical barrier between the blood and the basement membrane, regulating vascular toning, and modulating the immune system [4]. SARS-CoV-2 viral attack occurs through binding of the viral spike (S) protein to endothelial human angiotensin-converting enzyme 2 (ACE2) receptor and host serine protease transmembrane protease, serine 2 (TMPRSS2) priming [5]. The viral attack directly causes membrane disruption and damage to endothelial cells or indirectly leads to host inflammatory effects [5,6]. Both phenomena lead to endothelial dysfunction (i.e., endotheliitis, endothelialitis, and endotheliopathy) [3], characterized by endothelial activation and decreased endothelium-dependent vasodilation, hyperpermeability, and inflammation/leukocyte adhesion, resulting in a proinflammatory, procoagulant, and proliferative state.

The increased risk of severe illness in patients with existing cardiovascular diseases (CVDs) and those developing CVDs without pre-existing comorbidities following COVID-19 infection is strongly linked to high levels of proinflammatory cytokines (i.e., “cytokine storm syndrome”); severe local vascular dysfunction caused by the extension of widespread alveolar and interstitial inflammation to the pulmonary vasculature; and the intense oxidative stress associated with COVID-19 infection [7,8,9,10]. Ongoing symptomatic COVID-19 (present from 4 weeks and up to 12 weeks) and post-COVID-19 syndrome (present for >12 weeks and not attributable to alternative diagnoses) have been identified based on the available data [11]. Rezel-Potts et al. [12] evaluated the net long-term impacts of COVID-19 infection on cardiovascular and diabetes outcomes for a large population-based case-control study (428,650 COVID-19 cases vs. 428,650 controls). They reported a 6-fold increase in overall cardiovascular diagnoses, an 11-fold increase in pulmonary embolism, a 6-fold increase in atrial arrhythmias, and a 5-fold increase in venous thromboses among acute COVID-19 patients with no pre-existing CVDs in the first four weeks after infection. Moreover, these risks increased for up to 3 months following the infection [12]. Jung and colleagues [13] reported the existence of abdominal microcirculatory disorders in severe COVID-19 illness in contrast enhanced ultrasonography (CEUS). The 2022 study by Koutsiaris et al. also detected the 6-fold increase in microthrombosis using conjunctival video capillaroscopy (CVC) in COVID-19 survivors after having been discharged from the hospital [14]. Although the risk of developing new CVDs declined to baseline in patients without pre-existing CVDs, as reported by Rezel-Potts and colleagues [12], another observational study by Knight et al. [15] reported that the elevated risk of developing CVDs might persist for up to 49 weeks after COVID-19 infection.

The incident cardiovascular complications can be life-threatening in severe COVID illness. Because COVID-19’s symptoms and severity are varied, medical interventions must be carefully balanced between benefits and risks. A healthy nutritional status is one of the most important factors that supports immune homeostasis, particularly on COVID-19 days [16]. A deprived nutritional status due to low dietary quality could significantly contribute to an impaired immune system, especially since the continuous emergence of new variants constantly threatens vaccine-induced immunity [17]. A potential solution to health challenges in the COVID-19 era is enhancing patients’ general immunity via dietary intervention and nutraceutical supplementation. Omega-3 polyunsaturated fatty acids, also called n-3 polyunsaturated fatty acids (N3PUFAs), and their metabolites play an important role in the synthesis of various inflammatory mediators, such as prostaglandin (PG), leukotriene (LT), thromboxane (TX), and protectin.

The major causes of severe illness following COVID-19 infection are related to immune system overdrive leading to cytokine storms and, thereby, a potential cause of severe CVD. In 2022, a case-control study reported that lower N3PUFA intake was inversely associated with an increased likelihood of developing severe illness following COVID-19 infection after the adjustment of confounders [18]. Increasing N3PUFA intake in the diet or through supplementation could potentially promote better immune function and decrease the severity of the inflammatory response. As N3PUFAs are abundantly available in marine sources and have generally recognized as safe (GRAS) status [19], they could be a relatively safe and convenient prophylactic and conjunctive supplementation or treatment approach for patients who have comorbid CVD. In light of this, the current review aims to evaluate the role of N3PUFAs in antiinflammation and their potential health benefits in protecting cardiovascular health during and post-COVID-19 infection.

## 2. Risk of Cardiovascular Health Deterioration during and Post-COVID-19

### 2.1. Epidemiological Evidence: Endothelial Dysfunction Is Linked to COVID-19-Associated Cardiac Microthrombotic Dysfunction

SARS-CoV-2 comprises four structural proteins: spike (S), membrane (M), nucleocapsid (N), and envelope (E) proteins. After binding to ACE2, the S protein changes its shape through the endosomal pathway (also known as proteolysis) and helps the virus and host cell fuse so that viral RNA can be injected into the host cell [20]. In the endoplasmic reticulum (ER), SARS-CoV-2 proteins assemble with viral RNA to form virions, which are then released from the host cells. While 81% of symptomatic individuals had the relatively mild disease after infection, 14% developed severe disease with dyspnea, hypoxia, or lung involvement over 50% by imaging, and 5% developed critical disease with respiratory failure, shock, and/or multiorgan dysfunction [21]. Since the early phases of the COVID-19 pandemic, pathophysiological traits, including severe endothelial injury [5], alveolar capillary microthrombi [22], venous thromboembolism [23], systemic inflammatory response [24], and acute respiratory distress syndrome (ARDS) have been reported [25].

Particularly, CVD shares a bidirectional relationship with the severity of COVID-19. Patients hospitalized with severe COVID-19 have a significantly higher rate of incidents of clinical cardiovascular complications [26,27], encompassing acute heart failure, arrhythmias, venous thromboembolism, cardiogenic shock, arterial thrombosis, myocardial ischemia or infarction, acute stroke, and myocarditis. Early studies in 2020 reported a high incidence of venous thrombosis (33%) [28] and pulmonary embolism (20.6%) [29] among severe COVID-19 illness, while around 50% of patients who required intensive care had thrombosis [30,31,32,33]. The retrospective cohort study by Argenziano et al. [26] reported that during the early pandemic stage in New York, the most common clinical complications among the 1000 patients studied encompassed over 25% with thromboembolism and around 20% with arrhythmias and heart failure. Similarly, a significantly higher relative risk (RR) (up to 25%) of developing thrombotic complications was reported in the Netherlands after COVID-19 infection, in particular venous thromboembolism, when compared to influenza infection (11% increased RR) [34]. Numerous population-based studies also provided strong evidence that individuals with pre-existing cardiometabolic disease were prone to develop severe COVID-19 illness and a higher risk of life-threatening complications [35,36]. This could be potentially related to pre-existing subclinical pathophysiological abnormalities, such as endothelial dysfunction, procoagulation (platelet hyperreactivity and propensity to coagulopathy), and dysfunctional immune responses [37]. Furthermore, the invasion by SARS-CoV-2 of the endothelium of blood vessels can lead to direct endothelial damage and triggers a marked immune response that further causes additional endothelial dysfunction [5,38].

This endothelial dysfunction may continue after COVID-19. A 2022 population-based study in Sweden quantified the increased risk of thromboembolic events among individuals who recovered from COVID-19 compared to controls without COVID-19 but matched for comorbidities and other risk factors [39]. The study reported that incidence rates were significantly increased for a first deep vein thrombosis up to 70 days after COVID-19, pulmonary embolism up to 110 days, and a bleeding event up to 60 days [39]. More studies have reviewed the potential causal factors that contributed to the elevated cardiovascular risks, which are directly linked to immunothrombosis, not only in acute but also post-COVID-19 infections.

### 2.2. Prothrombotic State in COVID Infection

It has been hypothesized that the endothelial dysfunction caused by SARS-CoV-2 infection can lead to a prothrombotic state that encompasses increased immunothrombosis and coagulation and decreased fibrinolysis [40] (Table 1).

Immunothrombosis is characterized as a mechanism by which monocytes and neutrophils activate the coagulation cascade as the host’s immune response fights the viral invasion. The uncontrolled release of proinflammatory cytokines and chemokines responsible for the innate immune system, including interleukin (IL)-6, IL-8, interferon (IFN)-γ, and IL-2, can result in an increase in platelet production and activity. IL-6 also increases the expression of TF on monocytes and endothelial cells to further worsen under failure dysfunction [41]. IFN-γ can also impair the vascular endothelium to promote a prothrombotic effect, whereas IL-2 upregulates plasminogen activator inhibitor-1 (PAI-1) to reduce fibrinolysis [42]. The elevated IL-8 attracts neutrophils to the infected cells, increasing the likely part of the formation of neutrophil extracellular traps (NETs) [41], which may lead to platelet-dependent NET-driven thrombosis (Table 1).

Platelet activation and recruitment to the site of endothelial injuries produces a platelet plug and acts as an adhesion site for coagulation factors. Simultaneously, the activated platelets secrete bioactive compounds (e.g., polyphosphates and other coagulation factors and immunological mediators (e.g., complement factors) to further promote the activation of coagulation factors by inhibiting tissue factor pathway inhibitor (TFPI) and promoting fibrin polymerization [40].

The direct invasion by SARS-CoV-2 into endothelial cells damages the intercellular junctions of the endothelium to expose the prothrombotic subendothelial matrix collagen, which activates the intrinsic coagulation pathway and tissue factor (TF), which then activates the extrinsic coagulation pathway. Both pathways activate factor X to cleave prothrombin to form thrombin, which cleaves fibrinogen to increase the amount of fibrin strands, stabilize platelet aggregates, and form a thrombus. A recent discovery of mechanisms linking SARS-CoV-2 infection to platelet hyperreactivity indicated that the spike protein on the envelope of virions or on the surface of SARS-CoV-2-infected cells could activate calcium-dependent chloride channels and scramblase transmembrane protein 16F (TMEM16F, also known as anctamin 6) expressed in platelets to potentially activate platelet adhesion and aggregation [40] (Table 1).

The viral infection can also induce vascular injury leading to pyroptosis, programmed cell death that releases various damage-associated molecular patterns (DAMPs) and pathogen-associated molecular patterns (PAMPs) [43]. The recognition of PAMPs by monocytes’ toll-like receptors (TLRs) and CD14 receptors stimulates the transcription and expression of TF [43]. The cumulative response to the DAMPs and PAMPs is likely to lead to the accumulation of immune cell expression of prothrombotic proteins through inflammation and contribute to hypercoagulation [40].

Blockage of the ACE2 receptor by SARS-CoV-2 viruses can also lead to increasing angiotensin II, which promotes the endothelial expression of an array of prothrombotic proteins, including P-selectin, TF, and von Willebrand factor (VWF). The raised combination of soluble thrombomodulin (i.e., endothelial glycoprotein) and the other prothrombotic proteins can directly activate the extrinsic coagulation pathway. An increase in inflammation in endothelial cells invaded by SARS-CoV-2 can induce TF expression on macrophages and platelets, impair the TFPI, which inhibits the TF pathway, and further induce coagulation [44].

## 3. Beneficial Potential of N3PUFAs as Conjunctive Supplements for Cardiovascular Health in Acute and Post-COVID-19

Although the mechanisms responsible for the beneficial effect of N3PUFAs on reducing CVD risk during and post-COVID infection have not been comprehensively established, a collection of research has provided insights into molecular mechanisms. It has been demonstrated that N3PUFAs have antithrombotic properties due to their ability to reduce platelet aggregation and adhesion, thereby preventing the formation of blood clots. Furthermore, N3PUFAs have shown to benefit cardiovascular health owing to their ability to decrease the production of proinflammatory molecules, which contribute to thrombosis, and increase the production of anti-inflammatory molecules, which aid in preventing thrombosis.

### 3.1. N3PUFA and Endothelial Function

N3PUFA has been shown to improve endothelial function through various mechanisms. The most significant effect of N3PUFA on endothelial cells is to increase nitric oxide (NO) vasodilation availability, which is mediated by NO synthase (eNOS) activation [45], and enhance eNOS activity [46]. N3PUFA may mitigate the vasoconstrictive effect of endothelin-1. Experiments conducted on human endothelial cells in cultures exposed to eicosapentaenoic acid (EPA) indicated a decrease in endothelin-1 production independent of the effects of EPA on NO [47]. Nevertheless, fish oil containing EPA in healthy subjects did not affect plasma endothelin-1 levels over a brief period [48], potentially due to the lower EPA dose. N3PUFA also demonstrated regenerative properties on the endothelium of the blood vessels that are mediated by the stimulation of endothelial progenitor cells in healthy individuals and patients with a high CVD risk [49] (Table 1).

N3PUFA may also influence the activity of endothelium-mediated vasodilation and vascular smooth muscle cell (VSMC) function by stimulating the release of adiponectin with vasodilatory properties from perivascular adipocytes. It has been demonstrated that N3PUFA increased plasma adiponectin levels in subjects taking oral supplements of fish oil [50]. Randomized controlled trials (RCTs) examining the association between dietary N3PUFA and endothelium-dependent vasodilation remain inconclusive. However, while several RCTs did not establish an improvement in vasodilation [51,52,53], two studies have shown the beneficial effects of N3PUFA on triglyceride concentration and inflammatory markers [53,54] (Table 1).

### 3.2. N3PUFA and Immunothrombosis

Increasing evidence has supported the antithrombotic properties of N3PUFA. A 2018 epidemiological study observed that a diet including a high intake of marine N3PUFAs (≥4.7 g/week) was associated with a reduced risk of venous thromboembolism (VTE) following unprovoked index events [55]. Again, a meta-analysis of prospective cohort studies confirmed N3PUFA’s protective effect on VTE and recurrent VTE [56]. The 2020 meta-analysis by Zhang et al. [56] evaluated its overall multi-variable adjusted RR, reporting the significant inverse association between N3PUFA consumption and the risk of venous thromboembolism (VTE) (RR = 0.89, 95% CI: 0.80–0.98, *p* = 0.024). The 2017 RCT by Bonutti and colleagues examined the antithrombotic effect of the daily consumption of 325 mg aspirin and 1 g N3PUFA-rich fish oil for 90 days [57]. Significant reduction in the incidence of VTE after surgery for primary total knee arthroplasty was reported [57]. A more recent RCT comparing the daily supplementation of 1g N3PUFA also reported significantly reduced deep vein thrombosis after 30 days [58].

There are several mechanisms by which N3PUFAs can exert their antithrombotic effects (Table 1). One of the most well studied mechanisms is their ability to inhibit platelet-activating factor and other prothrombotic pathways [59]. Similar to other mammalian cell membranes, the platelet membrane has a lipid bilayer consisting of an outer leaflet of choline-containing phospholipids (primarily phosphatidylcholine and sphingomyelin) and an inner leaflet of negatively charged aminophospholipids (i.e., phosphatidylethanolamine (PE) and phosphatidylserine (PS)) [60]. Numerous aspects of platelet function rely heavily on the platelet phospholipid membrane [60]. Several enzymes dependent on calcium and adenosine triphosphate (ATP) control the asymmetric distribution of membrane phospholipids [61]. Upon platelet activation, this asymmetric orientation of membrane phospholipids is disrupted, leading to calcium-dependent exposure of PS on the platelet surface. It is well known that PS surface exposure is a crucial component of normal hemostasis because it facilitates platelet procoagulant functions. The formation of the prothrombinase complex on the surface of platelet membranes facilitates the conversion of prothrombin to thrombin, thereby enhancing platelet activation [62,63].

Elevated levels of N3PUFAs may alter platelet phospholipid membrane composition and platelet function, thereby influencing the progression and thrombotic complications of CVD (Table 1). EPA and docosahexaenoic acid (DHA) can act on the platelet membrane to reduce platelet aggregation and thromboxane (TX) release via cyclooxygenase-1 (COX-1) and 12-lipoxygenase (12-LOX), which metabolize fatty acids into a group of beneficial oxylipins in platelets that significantly regulate platelet function in hemostasis and thrombosis [64]. Even in healthy subjects, four-week EPA supplementation successfully decreased platelet activation, an early step in platelet aggregation [65]. Particularly, as a COX substrate, it appears EPA is more active than DHA in altering platelet function, although the evidence is limited, while DHA appears to reduce the affinity of thromboxane A2 (TxA2) and increase the production of prostacyclin (PGI2) [64]. TXA2 is a proinflammatory molecule that promotes platelet aggregation and vasoconstriction, while PGI2 is an anti-inflammatory molecule that promotes vasodilation and inhibits platelet aggregation. By reducing the production of TXA2 and increasing the production of PGI2, N3PUFA can help prevent blood clot formation. In platelet hyperactivity prothrombotic conditions, a higher dose of N3PUFA might be necessary to reduce platelet activation and aggregation [64]. In addition, N3PUFAs have been shown to decrease the levels of fibrinogen, a protein involved in blood clot formation, and increase the levels of tissue plasminogen activator (tPA), an enzyme that breaks down blood clots. By reducing fibrinogen levels and increasing tPA levels, N3PUFAs can help to prevent the formation and promote the dissolution of blood clots [64].

These anticoagulant properties of N3PUFAs suggest possible effects on platelet aggregation in patients with severe COVID-19 illness. We can only speculate at this time as to whether N3PUFAs can mitigate the coagulopathy associated with severe COVID-19. More research is needed to fully understand the mechanisms by which N3PUFAs exert their antithrombotic effects and to determine the optimal dose and duration of N3PUFA supplementation for preventing thrombosis.

### 3.3. N3PUFA and Inflammation

During viral infection, inflammation induced by a relatively low level of oxidative stress (a low or medium concentration of reactive oxygen species (ROS) and reactive nitrogen species (RNS)) can be sufficient to generate an environment hostile enough to eradicate phagocytosed viruses (e.g., by pulmonary alveolar macrophages). Nonetheless, a prolonged or unresolved inflammatory process is associated with the overproduction of ROS, which compromises the antioxidant activities of the body’s defences, resulting in extensive cellular and tissue damage [66,67]. SARS-CoV-2 infection causes a significant increase in systemic oxidative stress and inflammation [68,69,70] (Table 1).

Moreover, N3PUFAs are essential in regulating lipid rafts and influencing cell membrane fluidity [71]. They are capable of incorporating themselves into the phospholipid bilayers of the cell membranes of neutrophils, an essential part of the innate immune system, and produce a range of lipid mediators with hormone-like actions (including prostaglandins, leukotrienes, and maresins) [71], primarily targeting the sites of tissue damage and infections. N3PUFA also improves the function of macrophages by provoking major alterations in gene regulation to regulate the production and secretion of cytokines and chemokines, blunting M1 macrophage polarization and promoting M2 polarization, ultimately promoting phagocytosis [71]. Other studies have reported the anti-inflammatory mechanisms of N3PUFA, including in downregulating nuclear factor-κ beta (NFκB), a transcription factor involved in cell signaling that initiates an inflammatory response by the innate immune system [72,73]; inducing interferons (IFNs) that inhibit viral replication [74]; and affecting the motility of CD4+ [75] and CD8+ T cells [76] and modifying their ability to reach target tissues, thereby potentially modulating cytokine responses to viral attack.

N3PUFAs are metabolic substrates of lipoxygenase and cyclooxygenase, which produce “specific pro-resolving mediators” ((SPMs); e.g., resolvins, protectins, and maresins) that end acute inflammatory responses. EPA competes with inflammatory arachidonic acid (AA), the metabolic precursor of proinflammatory and prothrombotic eicosanoids in cell membrane phospholipids, for the eicosanoid enzymatic route, reducing prostaglandins, leukotrienes, and thromboxanes [77] (Table 1).

N3PUFA SPMs reduce proinflammatory cytokines, leukocyte migration, and macrophage activity at the inflammatory site [78]. If healthy participants ingest more than 2 g/day of N3PUFAs, their primary mononuclear cells release less tumor necrosis factor-α (TNF-α), interleukin-1, and interleukin-6 during endotoxin stimulation [79]. 3NPUFAs also suppress interleukin-2 and T-lymphocyte proliferation, adhesion molecules, and platelet-activating factor’s prothrombotic action [79] (Table 1).

Activator protein 1 (AP-1) and NFκB are downregulated by 3NPUFA binding to G-protein receptor 120 [80]. NFκB stimulates proinflammatory cytokines and adhesion molecules, while AP-1 activates TNF-a. Lastly, N3PUFA influences inflammasomes, the innate immune system sensor and receptor proteins that form intra-cytoplasmic complexes in response to damaging stimuli. Infection, wounded cell host molecules, and cardiovascular risk factors such as advanced glycation end products and oxidized low-density lipoproteins activate inflammasomes. EPA and DHA can suppress NFκB and stimulate inflammasome lysosomal autophagy [72] (Table 1).

## 4. N3PUFA Form and Bioavailability

Since 1994, N3PUFA has been investigated and credited for its health benefits; among the most tremendous is in improving cardiovascular health. N3PUFAs have conferred cardiovascular benefits by reducing triglycerides, anti-inflammation, vasodilation, antihypertension, and platelet aggregation and by improving endothelial function [81]. N3PUFA was named based on the presence of the closest double bond to the methyl end of the hydrocarbon (acyl) chain being on carbon number three, counting the methyl carbon as number one. Within the N3PUFA family of polyunsaturated fatty acids, the most well studied is linolenic acid (LNA) and its derivatives, including the plant-derived α-linolenic acid (ALA; 18:3n-3) and EPA (20:5n-3), DHA (22:6n-3) [15], and docosapentaenoic acid (DPA; 22:5n-3) which are derived from marine sources [82] (Figure 1). The available in vivo studies and RCTs have focused on EPA and DHA, hence, this review will mainly discuss these two forms of N3PUFAs due to the availability of the data.

The metabolic conversion pathway of plant-derived ALA to bioactive EPA requires the help of Δ6-desaturase to form stearidonic acid, which then transforms to eicosatetraenoic acid via elongation, then to DHA by desaturation with the involvement of Δ5-desaturase. In humans, the metabolism is influenced by various factors (e.g., age, sex, hormonal change, genetics, etc.), the conversion rate can be relatively low, and the health benefits of ALA are limited [83]. Only around 8% of dietary ALA is converted to EPA and less than 4% to DHA in healthy young males, whereas in healthy young females, 21% of dietary ALA is converted to EPA and 9% to DHA [84,85]. Moreover, ALA has low bioavailability due to a higher rate of oxidation. In comparison, DHA has more bioavailability owing to its characteristic as a poor β-oxidation substrate [16]. Marine-derived EPA and DHA are considered better N3PUFA sources.

Following ingestion, N3PUFA is hydrolyzed, like other dietary lipids, in the intestines to free fatty acids (FFAs) and monoglycerides, which can then be incorporated into micelles after bile salt emulsification and absorbed into enterocytes by passive diffusion into chylomicrons [86]. The chylomicrons containing FFAs are delivered to various organs through lymphatic circulation for further metabolism [86]. Absorption and thus, bioavailability, is affected by factors such as intestinal pH, bile secretion, microorganisms, the type of chemical bond, concurrent food consumption, other components such as calcium [83], and the different forms of N3PUFA. N3PUFA exists in various forms, including as FFAs (e.g., free EPA, DHA, and DPA), triglycerides (TGs), ethyl esters (EEs), and phospholipids [84,86]. Unrefined fish oil contains mainly triglycerides with various amounts of N3PUFAs (i.e., EPA, DHA, and DPA as fatty acids) attached to glycerol in low concentrations.

Various purification methods have been adopted to increase the EE and TG forms of N3PUFAs. Among all methods, the ethylating purification method removes the glycerol backbone of triglycerides to release EPA and DHA, while removing short-chain fatty acids. FFAs are then esterified to an ethanol backbone to form ethyl esters (i.e., the EE form N3PUFA). The alternative method to increase N3PUFA content is to break down concentrated EE forms of N3PUFAs into FFAs and then esterify the FFAs to a glycerol alcohol backbone to form re-esterified TG forms of N3PUFA [87]. As the EE form requires an extra hydrolysis step to separate the FFAs from the ethyl carrier in human intestines [88], the FFAs and TG forming N3PUFAs have more bioavailable esters. In an acute study, the absorption of EPA in its TG form was 90% and 60% in the EE form [87]. A two-week study of 72 adults [89] reported a higher bioavailability of 3.3 g of re-esterified TG from EPA and DHA (124%) compared to natural fish oil, while the EE form (73%) was lower. The significantly lower bioavailability of EE forms compared to TG forms tended to be of a short duration (8–12 h) and ensured that a large dose of N3PUFAs (over 3 g of EPA and DHA) was provided to participants. However, the results from long-term comparative studies suggest no significant difference in the absorption of EPA and DHA between the TG and EE forms when N3PUFA is routinely consumed as a dietary supplement. A long-term study [90] reported no significant difference in bioavailability between the TG and EE forms after nine healthy males consumed 1.1 g EPA and 0.37 g DHA over a three-month period. Sadovsky and Kris-Etherton [91] pointed out that the beneficial effects of the EE form on objective health parameters, such as decreasing plasma triglycerides, initiated 1 month post-supplementation and reaching maximum effectiveness at 2 months. However, a short-term statistically significant difference in absorption and bioavailability does not necessarily reflect the overall bioavailability and clinical impact in the long term.

## 5. A High N3PUFA Dose Can Be Essential in Protecting Cardiovascular Health from COVID-19

N3PUFA has conferred cardiovascular health by reducing inflammation, oxidative stress, improving arterial and endothelial functions, and reducing platelet aggregation [81]. Since our bodies cannot synthesize N3PUFAs, we highly rely on dietary intake to replenish them [85]. The epidemiological evidence has reported that intake of EPA and DHA from the diet is strongly associated with fatty fish consumption. In contrast, the intake of N3PUFA varies significantly among different populations and is generally lower than the recommended 0.2–0.5 g/day for general adults [92,93,94] (depending on the various authorities making the dietary recommendation guidelines) in most Western countries of which the main protein source is meat instead of fish [95,96]. The inclusion of supplements that contain EPA and DHA is essential if the daily recommendation cannot be met through food intake only.

Particularly, the well known biological parameter, triglyceride concentration, has demonstrated a dose-dependent relationship with N3PUFAs. A significant reduction (20–50%) in blood TG was reported in patients with high baseline TGs after consuming 3 to 4 g/day of EPA or a combination of EPA and DHA [97]. However, controversies surrounding the clinical trials involving various N3PUFA daily doses emphasize the importance of high-dosage N3PUFAs in reducing CVD risks, including the combined stroke, MI, and death from CVD causes and major cardiovascular events. Therefore, to establish the dose that can demonstrate a clinically significant cardiovascular benefit, previous RCTs that studied the association between N3PUFA and CVD have been reviewed. Although there is a lack of consensus among scientists and clinicians, clear evidence from decades of studies is able to support recommendations.

Since the first landmark clinical trial that investigated the cardiovascular protective effect of N3PUFA in 1999, controversies have been reported. A GISSI-P study of an Italian population was the first study that demonstrated that 1 g of N3PUFA (a combination of EPA and DHA) supplementation per day significantly reduced the RR of death by 10% (95% CI: 1–18%) and severe cardiovascular events by 17% (95% CI: 3–29%), compared to the control group that consumed 300 mg/day of vitamin E [98]. A later GISSI-HF study again demonstrated a significant reduction in the hazard ratio (HR) of death and hospital admissions for cardiovascular reasons after subjects were on the same dose for 3.9 years [99]. The fundamental development of CVD treatment has been achieved, including aggressive therapy, since this GISSI-P study was published. Yokoyama et al. [99] also demonstrated in the 2007 JELIS study that the daily addition of 1.8 g EPA to standard statin medication per day significantly reduced the related risks of major coronary events in Japanese subjects who had equal to or higher than 6.5 mmol/L total cholesterol after a five-year follow-up. However, in the OMEGA study, where the majority of subjects received statin therapy at baseline, there was no significant improvement in sudden cardiac death, total mortality, or major adverse cerebrovascular and cardiovascular events (MACEs) [100]. There is a possibility that the pre-clinical trial optimal medical therapy could have contributed to this insignificant efficacy (Table 2).

Notably, the study populations in large-scale studies that showed a significant reduction in CVD risk were from high seafood intake regions, such as the Italian populations in the GISSI-P and GISSI-HF studies and the Japanese population in the JELIS study, who potentially had higher baseline N3PUFA concentrations due to their high dietary supply (Table 1). Previous studies have raised the possibility that a threshold of endogenous levels may be required to show the statistical significance of N3PUFA on CVD risk. Populations with a higher baseline of N3PUFA reserves may need a lower dose of N3PUFAs to show statistically significant improvement in their cardiovascular health; on the contrary, Western populations with lower fatty fish consumption are likely to require a higher N3PUFA dose. A 2010 small RCT of elderly Norwegian males (*n* = 563) at high risk of developing CVD (72% without overt CVD), the DOIT study, reported a tendency towards reduction in all-cause mortality and cardiovascular events that reaching statistical significance after the subject was on a doubled dose (2.4 g/day) despite the smaller sample size [102]. The VITAL study [110] also showed that the subjects, who received the most cardiovascular benefit from a N3PUFA supplement, had the lowest baseline levels of N3PUFA concentration (Table 2).

The findings from later studies using low-dose N3PUFAs (0.376–1 g/day) failed to demonstrate its sufficiency for populations who had a lower average of fatty fish consumption to reach the same therapeutic benefit to lower cardiovascular risk [100,103,104,105,106,107,109,110] (Table 1), including the SU.FOL.OM3 study, Alpha-OMEGA study, OMEGA study, ORIGIN study, R and P study, AREDS-2 study, and ASCEND study. Although the VITAL study of the US cohort found that daily administration of 2000 IU/day of vitamin D3 and 1 g/day of N3PUFAs (a combination of EPA and DHA) did not significantly reduce the risk of CVD when compared to the placebo group after 5.3 years of intervention [110], a statistically significant decrease in the HR of MI was reported. It was not until the OMEGA-REMODEL study found that high-dose N3PUFAs (4 g/day of an EPA and DHA combination) appeared to be beneficial up to 6 months after acute MI, that a reduction in adverse left ventricular remodeling, non-infarct myocardial fibrosis, and serum biomarkers of systemic inflammation beyond the current guideline-based standard of care were demonstrated [108]. The most recently conducted multi-center REDUCE-IT study reported that a high N3PUFA dose of 4 g/day (icosapent ethyl, highly purified EPA) significantly reduced major cardiovascular events in a multi-population, particularly in US participants who had a low baseline N3PUFA level [111] (Table 2). The treatment cohort significantly reduced the primary endpoint (a composite of CVD death, non-fatal MI, non-fatal stroke, CV revascularization, or unstable angina) by 25% and the secondary endpoint MACE by 26% [111]. The sub-cohort of the US population experienced a reduced RR of all-cause mortality by 30% and absolute risk by 2.6% [111] (Table 2).

Recent meta-analyses have examined the potential sources of heterogeneity in the effects of N3PUFA on cardiovascular health. A 2019 meta-analysis by Hu and colleagues [112] conducted a meta-regression of 13 RCTs, excluding REDUCE-IT, and concluded that marine N3PUFA supplementation was negatively associated with the risk of MI (RR = 0.92, 95% CI: 0.86, 0.99; *p* = 0.020), CHD death (RR = 0.92, 95% CI: 0.86, 0.98; *p* = 0.014), total CHD (RR = 0.95, 95% CI: 0.91, 0.99; *p* = 0.008), CVD death (RR = 0.93, 95% CI: 0.88, 0.99; *p* = 0.013), and total CVD (RR = 0.97, 95% CI: 0.94, 0.99; *p* = 0.015). The negative association was further strengthened when the REDUCE-IT study was included [112]. Bernasconi and colleagues [113], in their updated 2020 meta-analysis, stated that N3PUFAs of an EPA and DHA combination statistically significantly reduced the risk of CVD and MI by 9% and 13%, respectively. Moreover, a dose-dependent association was reported between the reduction in MI risk (9% reduction) and an additional 1 g/day of N3PUFA [113], indicating that the higher dose provided significantly higher protection. One 2017 meta-analysis that reviewed the minimal dose required for a clinically meaningful change in triglyceride concentration suggested that the low dose of N3PUFA could explain the inconsistent results in previous RCTs (<1.5 g/day of an EPA and DHA combination) [114]. A recent update on the dose recommendation of N3PUFA reviewed the threshold of the baseline N3PUFA index (a calculated value to express the amount of systemic EPA and DHA in weight (%) present in the tissue cell membrane lipid fractions), and the suggested supplementation dose suggested of high-dose N3PUFAs (4 g/day) appeared to be more beneficial among people with low baseline N3PUFA (<8% N3PUFA index, a measurement of serum N3PUFA levels) [115]. In contrast, the low dose (1 g/day) only benefited people with a high baseline (≥8% N3PUFA index) [115]. Further work including clinical trials on high-dose concentrated ethyl ester (i.e., the EE form) N3PUFAs will be conducted and funded by Pharma New Zealand PNZ Limited (Hamilton, New Zealand).

## 6. Conclusions

COVID-19 can cause a hyperinflammatory response that leads to the formation of blood clots, which can affect blood vessels throughout the body, including those that supply the heart. There is growing evidence that COVID-19-related immunothrombosis can increase the risk of CVD. Patients with pre-existing CVD are at a higher risk of experiencing complications from COVID-19. It is vital for healthcare providers to monitor COVID-19 patients for signs of CVD and provide appropriate treatment to reduce the risk of complications.

At this time, there are no clear studies that demonstrate the positive effects of N3PUFA on COVID-19 patients. However, high dose concentrated N3PUFAs (4 g/day) have been shown to regulate and modulate certain negative immunological overreaction effects, limit coagulopathy, and influence cell signaling and gene expression. They are well known to have antithrombotic, anti-inflammatory, and pro-resolving properties, which can be advantageous for COVID-19 patients. The ingestion of N3PUFAs and/or their metabolites may prevent and manage cardiovascular and thrombotic issues in COVID-19 patients. It is, therefore, prudent to study the possible uses of fish oil/N3PUFA supplementation as an adjuvant to medication in COVID-19 patients at risk of vascular thrombotic events.

## Figures and Tables

**Figure 1 metabolites-13-00630-f001:**
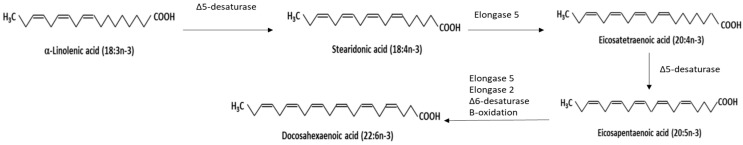
Different forms of N3PUFAs and the metabolic pathway of converting α-linolenic acid (ALA; 18:3n-3) to eicosapentaenoic acid (EPA; 20:5n-3) and docosahexaenoic acid (DHA; 22:6n-3).

**Table 1 metabolites-13-00630-t001:** Potential mechanism of antithrombotic effects of N3PUFA.

Pathological Significance	Effects of N3PUFA ^1^
Endothelial function	↑ NO ^2^ and eNOS ^3^ activity
	↓ endothelin-1
	↑ endothelium-mediated vasodilation
	↑ VSMC ^4^ relaxation
	↑ adiponectin

Immunothrombosis	↓ platelet activation
	↓ platelet aggregation and TX ^5^ release via COX-1 ^6^ and 12-LOX ^7^
	↓ affinity of TxA2 ^8^
	↑ PGI2 ^9^ production
	↓ fibrinogen level
	↑ tPA ^10^ level
	↓ proinflammatory cytokines (ILs ^11^, TNF ^12^)

Inflammation	↓ regulation of AP-1 ^13^ and NFκB ^14^
	↓ T-lymphocyte proliferation

^1^ Omega-3 polyunsaturated fatty acids. ^2^ Nitric oxide. ^3^ Nitric oxide synthase. ^4^ Vascular smooth muscle cell. ^5^ Thromboxane. ^6^ Cyclooxygenase-1. ^7^ 12-Lipoxygenase. ^8^ Thromboxane A2. ^9^ Prostacyclin. ^10^ Tissue plasminogen activator. ^11^ Interleukins. ^12^ Tumor necrosis factor-α. ^13^ Activator protein 1. ^14^ Nuclear factor kappa B. ↑ Increase ↓ Decrease

**Table 2 metabolites-13-00630-t002:** Characteristics of parallel-design RCTs of N3PUFA.

Year	Trial	Population	No. of Subjects	Age (years)	Male (%)	Subject Characteristics	N3PUFA ^1^, Dose	Control	Study Period	Result
1999	GISSI-P [98]	Italian	11,324	59	84.7	Surviving recent (≤3 months) myocardial infarction	N3PUFA, 1 g/day	Vitamin E, 300 mg/day	3.5 years	↓ RR ^2^ of death = 10% (95% CI: 1–18%); ↓ RR of CVD = 17% (95% CI: 3–29%)
2007	JELIS [99]	Japanese	18,645	Average 61	31.4	Total cholesterol ≥6.5 mmol/L	EPA ^3^, 1.8 g/day; statin	Statin only	5 years	↓ 19% RR in major cardiovascular events
2008	GISSI-HF [98,100]	Italian	955	≥18	77.8	With chronic heart failure of New York Heart Association classes II–IV, irrespective of cause, and left ventricular ejection fraction	N3PUFA, 1 g/day	Placebo	3.9 years	↓ HR ^4^ of death = 0.91 (95% CI: 0.833–0.998); ↓ HR of hospital admission for cardiovascular reasons = 0.92 (95% CI: 0.849–0.999)
2010	DOIT [101]	Norwegian	563	64–76	100	Without overt cardiovascular disease	N3PUFA, 2.4 g/day	Placebo (corn oil)	3 years	↓ HR of death = 0.57 (95% CI: 0.29–1.10); ↓ HR of cardiovascular events = 0.86 (95% CI: 0.57–1.38)
2010	SU.FOL.OM3 [102]	French	2501	45–80	79.5	With a history of myocardial infarction, unstable angina, or ischemic stroke	5-methyltetrahydrofolate, 560 μg/day; vitamin B6, 3 mg/day; vitamin B12, 20 μg/day; N3PUFA, 0.6 g/day	Placebo	4.7 years	No significant effect on major cardiovascular events
2010	Alpha-OMEGA [103]	Dutch	4837	60–80	78.0	Had a myocardial infarction, received state-of-the-art antihypertensive, antithrombotic, and lipid-modifying therapy	N3PUFA, 0.376 g/day (EPA, 0.226 g/day; DHA ^5^, 0.150 g/day)	ALA, 1.9 g/day	NA	→ HR of major cardiovascular events = 1.01 (95% CI: 0.87–1.17)
2010	OMEGA [104]	German	3851	64	74.4	3 to 14 days after acute myocardial infarction	N3PUFA (EE form), 1 g/day	Placebo	1 year	No significant difference in sudden cardiac death, total mortality, major adverse cerebrovascular and cardiovascular events
2012	ORIGIN [105]	Canadian	12,536	≥50	40.0	At high risk for cardiovascular events and had impaired fasting glucose, impaired glucose tolerance, or diabetes	N3PUFA (EE form), 0.9 g/day	Placebo	6.2 years	→ HR of time to death or admission to the hospital for cardiovascular causes, 0.97 (95% CI: 0.88–1.08)
2013	R and P [106]	Italian	12,513	≥65	61.5	with multiple cardiovascular risk factors or atherosclerotic vascular disease but not myocardial infarction	N3PUFA (EE form), 1 g/day	Placebo	1 year	→ HR of the rates of major cardiovascular events, 1.01 (95% CI: 0.93–1.10)
2014	AREDS-2 [107]	American	4203	50–85	56.8	With stable, existing CVD (>12 months since initial event)	N3PUFA, 1 g/day (EPA, 650 mg/day; DHA, 350 mg/day); lutein, 10 mg/day; zeaxanthin, 2 mg/day	Placebo	4.8 years	→ HR of risk of CVD or secondary CVD outcomes, 0.95; 95% CI, 0.78–1.17
2016	OMEGA- REMODEL [108]	American	358	>21	65.0	With an acute MI	N3PUFA, 4 g/day (EPA, 465mg/day; DHA, 375 mg/day)	Corn oil (linoleic acid, no N3PUFA, 600 mg/day)	6 months	↓ LVESVI ^6^ (–5.8%, *p* = 0.017); ↓ Non-infarct myocardial fibrosis (−5.6%, *p* = 0.026)
2018	ASCEND [109]	British	15,480	≥40	62.6	With diabetes but without evidence of atherosclerotic cardiovascular disease	N3PUFA, 1 g/day	Olive oil, 1 g/day	7.4 years	No significant difference in serious vascular event or revascularization
2019	VITAL [110]	American	25,871	>50 (males) >55 (females)	49.9	Healthy	N3PUFA, 1 g/day; vitamin D3 2000 IU/day	Placebo	5.3 years	No significant difference in serious vascular event; ↓ HR of MI = 0.71 (95% CI:0.59–0.9)
2019	REDUCE-IT [111]	71% (US, Canada, Netherlands, Australia, New Zealand, and South Africa), 25.8% (Eastern European), 3.2% (Asia-Pacific)	8179	≥45 (established CVD) ≥50 (established T2DM)	71.2	With established cardiovascular disease or with diabetes and other risk factors, receiving statin therapy, fasting triglyceride level of 135 to 499 mg per deciliter (1.52 to 5.63 mmol per liter), a low-density lipoprotein cholesterol level of 41 to 100 mg per deciliter (1.06 to 2.59 mmol per liter)	EPA (icosapent ethyl highly purified EPA formulation), 4 g/day	Placebo	4.9 years	↓ HR of major cardiovascular events = 0.75 (95% CI: 0.68–0.83)

^1^ N3PUFA, n-3 polyunsaturated fatty acid; ^2^ RR, relative risk; ^3^ EPA, eicosapentaenoic acid; ^4^ HR, hazard ratio; ^5^ DHA, docosahexaenoic acid; ^6^ LVESVI, left ventricular systolic volume index. ↑ Statistically significant increase ↓ Statistically significant decrease → No statistically significant change

## Data Availability

Data is not publicly available due to privacy.

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
