# Peer review of "Potential Benefits of Omega-3 Polyunsaturated Fatty Acids (N3PUFAs) on Cardiovascular Health Associated with COVID-19: An Update for 2023"

_metabolites, 2023, doi:10.3390/metabo13050630_

Round 1

Reviewer 1 Report

Lines 45-48 are not clear and should be rephrased

Line 125-CVD should be used as an abbreviation, as it has already been introduced earlier 

Line 196-"sell" should be "cell" and "thatch" should be "that"

Line 212-I think that a short paragraph should be introduced about

N3PUFA in general, and especially the ones that are described later on, before writing about their effects and mechanisms of action

Line 213-the sentence is not clear

Line 219-it should be emphasized that fish oil contains EPA

Table 1 should be placed after subsection 2.2.

Line 242-245/the sentence is hard to understand and should be rephrased i.e. divided into two sentences.

Where Table 1 is mentioned in the text, the fullstops should be after brackets

Line 284-n-3PUFA should be N3PUFA

Author Response

Response to editors and reviewers:

We would like to thank the reviewers and the editors for their valuable insights and constructive critique. We have revised the manuscript to address the reviewers’ comments with the editor’s approval. Please find below our response to the reviewers in blue. The changes have been made in the version 2 manuscript.

Reviewer #1:

Lines 45-48 are not clear and should be rephrased

RESPONSE: We have rephrashed the line 45-48.

“SARS­CoV-2 viral attack occurs through binding of the viral spike (S) protein to endothelial human angiotensin-converting enzyme 2 (ACE2) receptor and, facilitated by host serine protease Transmembrane protease, serine 2 (TMPRSS2) priming [5]. The viral attack,  directly causes membrane disruption and damage to endothelial cells or in-directly leads to host inflammatory effects [5].”

Line 125-CVD should be used as an abbreviation, as it has already been introduced earlier

RESPONSE: CVD is used as an abbreviation as suggested by reviewer. The context has been changed accordingly.

“Particularly, cardiovascular diseaseCVD shares a bidirectional relationship with the severity of COVID-19.”

Line 196-"sell" should be "cell" and "thatch" should be "that"

RESPONSE: The sentence has been corrected accordingly.

“The viral infection can also induce vascular injury leading to pyroptosis, which is a program to csell deaths thatch release various damage-associated molecular patterns (DAMPs) and pathogen-associated molecular patterns (PAMPs) [47].”

Line 212-I think that a short paragraph should be introduced aboutN3PUFA in general, and especially the ones that are described later on, before writing about their effects and mechanisms of action

RESPONSE: A short paragraph has been added to introduce N3PUFA in general.

“Although the mechanisms responsible for the beneficial effect of N3PUFAs on re-ducing CVD risk during and post COVID infection have not been comprehensively established, a collection of research has provided insights of molecular mechanisms. It has been demonstrated that N3PUFAs have antithrombotic properties due to their ability to reduce platelet aggregation and adhesion, thereby preventing the formation of blood clots. Furthermore, N3PUFA have shown to benefit the cardiovascular health owing to their ability to decrease the production of pro-inflammatory molecules, which contribute to thrombosis, and increase the production of anti-inflammatory molecules, which aid in preventing thrombosis.”

Line 213-the sentence is not clear

RESPONSE: The sentence has been edited.

“N3PUFA has been shown to reduce the improve damaged endothelial function through various mechanisms.”

Line 219-it should be emphasized that fish oil contains EPA

RESPONSE: The sentences have been edited.

“Experiments conducted on human endothelial cells in cultures exposed to eicosapentaenoic acid (EPA)indicated a decrease in endothelin-1 production independent of the effects of EPA on NO [51]. Nevertheless, fish oil containing EPA in healthy subjects did not affect plasma endothelin-1 levels over a brief period [52], potentially due to the lower EPA dose.”

Table 1 should be placed after subsection 2.2.

RESPONSE: Table 1 has been moved to subsection 2.2.

Line 242-245/the sentence is hard to understand and should be rephrased i.e. divided into two sentences.

RESPONSE: The sentences have been rephrased.

“The 2017 RCT by Bonutti and colleagues examined the anti-thrombotic effect of the daily consumption of 325 mg aspirin and 1g N3PUFA-rich fish oil for 90 days [61].  Significantly reduced reduction of the incidence of VTE after surgery for primary total knee arthroplasty was reported [61].”

Where Table 1 is mentioned in the text, the fullstops should be after brackets

RESPONSE: The fullstops have been moved.

Line 284-n-3PUFA should be N3PUFA

RESPONSE: “n-3 PUFA” has been changed to “N3PUFA”.

“These anticoagulant properties of n-3 N3PUFAs suggest possible effects on plate-let aggregation in patients with severe COVID-19 illness. We can only speculate at this time as to whether n-3 N3PUFAs can mitigate the coagulopathy associated with severe COVID-19.”

Reviewer 2 Report

This is a 2023 update-review work for the potential benefits of Ω3PUFAs on cardiovascular health that concludes on a very interesting proposition of using Ω3PUFAs as adjuvant treatment for COVID-19. However, there are important points that need to be addressed.

MAJOR POINTS

1] Title: the word “omega” is used but the acronym inside the parenthesis is N3PUFA. It would be more consistent to use Ω3PUFA. This holds for the rest of the text, but in the introduction, it could be explained that Ω3PUFA is the same as N3PUFA.

2] Abstract, Line 25: the “linearity” in the dose-response relationship has not been demonstrated in the manuscript. From the data presented on pages 12 and 13 the authors could try to make a diagram of HR versus dose (or RR versus dose) in order to demonstrate linearity, or if this is not possible, to demonstrate some kind of negative correlation between cardiovascular risk and Ω-3 dose.

3] Introduction-Line 65 & Section 2.1: The authors miss the opportunity to refer to research on human microcirculatory thrombosis in-vivo due to COVID-19 (Jung EM et al, Clin. Hem. Micr., 74 (2020):353–361; Koutsiaris et al, Clin. Hem. Micr., 82 (2022): 379–390).

Jung et al (2020) demonstrated, abdominal microcirculatory disorders in severe cases of COVID-19 infection (at the bedside in intensive care units), using contrast enhanced ultrasonography (CEUS).

Koutsiaris et al (2022) demonstrated, microthrombosis in COVID-19 survivors, shortly after hospital exit, using Conjunctival Video Capillaroscopy (CVC). The “5-fold increase in venous thromboses” in Line 65 of the manuscript, agrees very well with the 6-fold increase in microthrombosis at the smallest eye microvessels reported by Koutsiaris et al (2022) using the newly introduced microthrombosis index of POV.

In the title of Section 2.1 the word “microthrombosis” could be added after the last word “dysfunction”.

4] Introduction: Lines 84-97 should move to Section 3.1 “N3PUFA and inflammation”.

5] Section 4 on bioavailability should move after the Introduction Section since it gives important basic information on N3PUFA. A figure depicting the formulas of all kinds of Ω-3 would be nice. The authors should explain why they have concentrated their review on EPA and DHA.

6] Section 2.2: the connection of this section to Table 1 is not clear. Maybe a Table is missing. Table 1 connects clearly to Section 3.

7) Table 1: It would be clearer to make a different column for each pathological significance effect so that the effects of N3PUFA appear on 3 separate columns.

ADDITIONAL POINTS

8] Line 59: it is “syndrome” instead of “syndrom”.

9] Line 83: words are missing: “…and resolving…”.

10] Line 84: some references should be added on “cell membrane fluidity”.

11] Line 119: “…or over 50%...” improve syntax.

12] Line 172: put “oh” into the correct form.

13] Line 196: correct the phrase “…program to sell deaths…”.

14] Line 213: correct the phrase “…to reduce the improve…”.

15] The subtitles of sections 3.1, 3.2, and 3.3 should be the same as the entries of the pathological significance column of Table 1.

16] Line 236: how much was the high intake?

17] Line 238: correct to “meta-analyses”.

18] Line 290: change 3.2 to 3.3

19] Line 343: change “intestinal” to “intestine”. The acronym FFA should be explained in the first appearance.

20] Line 353: change into a new paragraph.

21] Line 361: improve syntax.

22] Line 379: it seems that “anti-oxidative” should read “oxidative”.

23] Line 397: improve syntax.

24] Table 2: Leave a blank line between studies to improve the readability in the “Result” column.

25] Line 403: HH should read HR.

26] Line 448: HA should read HR.

27] Line 460: “Table 1” should read “Table 2”.

28] Line 484: The “baseline N3PUFA index” should be defined. A citation is missing.

29] Line 506: in the phrase “oil/N3PUFA PUFAs” the “PUFAs” is redundant.

The quality of the English Language is good however, moderate editing will be required in some cases as it is described in the comments for authors.

Author Response

Response to editors and reviewers:

We would like to thank the reviewers and the editors for their valuable insights and constructive critique. We have revised the manuscript to address the reviewers’ comments with the editor’s approval. Please find below our response to the reviewers in blue. The changes have been made in the version 2 manuscript.

Reviewer #2:

1] Title: the word “omega” is used but the acronym inside the parenthesis is N3PUFA. It would be more consistent to use Ω3PUFA. This holds for the rest of the text, but in the introduction, it could be explained that Ω3PUFA is the same as N3PUFA.

RESPONSE: We thank reviewer#2 for the valuable suggestion. We have added the explanation in the introduction.

Line 19: “Accumulating literature demonstrates that omega-3 polyunsaturated fatty acid (n-3 polyunsaturated fatty acid, N3PUFA) can be incorporated into the phospholipid bilayer of cell membrane in the human body to positively affect the cardiovascular system, including improving epithelial function, decreasing coagulopathy, and attenuating uncontrolled inflammatory responses and oxidative stress.”

Line 81: “Omega-3 polyunsaturated fatty acids, also called n-3 polyunsaturated fatty acids (N3PUFA), and their metabolites play an important role in the synthesis of various inflammatory mediators, such as prostaglandin (PG), leukotrienes (LT), thromboxanes (TX), protectins, and resolving.”

2] Abstract, Line 25: the “linearity” in the dose-response relationship has not been demonstrated in the manuscript. From the data presented on pages 12 and 13 the authors could try to make a diagram of HR versus dose (or RR versus dose) in order to demonstrate linearity, or if this is not possible, to demonstrate some kind of negative correlation between cardiovascular risk and Ω-3 dose.

RESPONSE: We thank reviewer#2 for the valuable suggestion. We have deleted the word “linear” to improve the clarity of the abstract.

3] Introduction-Line 65 & Section 2.1: The authors miss the opportunity to refer to research on human microcirculatory thrombosis in-vivo due to COVID-19 (Jung EM et al, Clin. Hem. Micr., 74 (2020):353–361; Koutsiaris et al, Clin. Hem. Micr., 82 (2022): 379–390).

Jung et al (2020) demonstrated, abdominal microcirculatory disorders in severe cases of COVID-19 infection (at the bedside in intensive care units), using contrast enhanced ultrasonography (CEUS).

Koutsiaris et al (2022) demonstrated, microthrombosis in COVID-19 survivors, shortly after hospital exit, using Conjunctival Video Capillaroscopy (CVC). The “5-fold increase in venous thromboses” in Line 65 of the manuscript, agrees very well with the 6-fold increase in microthrombosis at the smallest eye microvessels reported by Koutsiaris et al (2022) using the newly introduced microthrombosis index of POV.

RESPONSE: We have updated the indruction and citation accordingly.

“Jung and colleagues [13] reported the existence of abdominal microcirculatory disorders in severe COVID-19 illness in contrast enhanced ultrasonography (CEUS). The 2022 study by Koutsiaris et al. also detected the 6-fold increase in microthrombosis using Conjunctival Video Capillaroscopy (CVC) in COVID-19 survivors after discharged from hospital [14].”

In the title of Section 2.1 the word “microthrombosis” could be added after the last word “dysfunction”.

RESPONSE: We have added “microthrombosis” to the title as suggested.

“2.1 Epidemiology evidence: endothelial dysfunction is linked to COVID-19 associated cardio microthrombosis dysfunction”

4] Introduction: Lines 84-97 should move to Section 3.1 “N3PUFA and inflammation”.

RESPONSE: We have moved the section to Section 3.3 “N3PUFA and inflammation”.

5] Section 4 on bioavailability should move after the Introduction Section since it gives important basic information on N3PUFA. A figure depicting the formulas of all kinds of Ω-3 would be nice. The authors should explain why they have concentrated their review on EPA and DHA.

RESPONSE: To better emphasise the review topic, the potential benefits of N3PUFA on cardiovascular health associated with COVID019, we decided to keep the bioavailability as a separate section instead of merging into the introduction. Therefore, the introduction would be focusing on the potential benefits of N3PUFA

We have added a figure (Figure 1) of all kinds of N3PUFA as suggested in section 4.

We have explained the reason that we have concentrated the review on EPA and DHA in section 4. “The available in vivo and RCTs are focusing on EPA and DHA, hence, this review will mainly discuss these two forms of N3PUFA due to data availability.”

6] Section 2.2: the connection of this section to Table 1 is not clear. Maybe a Table is missing. Table 1 connects clearly to Section 3.

RESPONSE: We have improved the connection between Table 1 and Section 2.2, including emphasing the platelet activation process.

7) Table 1: It would be clearer to make a different column for each pathological significance effect so that the effects of N3PUFA appear on 3 separate columns.

RESPONSE: We have added extra row between three pathological significance to improve the clarity.

ADDITIONAL POINTS

8] Line 59: it is “syndrome” instead of “syndrom”.

RESPONSE: We have corrected the typo to “syndrome”.

9] Line 83: words are missing: “…and resolving…”.

RESPONSE: We have corrected the sentence by deleting “and resolving”.

“Omega-3 polyunsaturated fatty acids, also called n-3 polyunsaturated fatty acids (N3PUFA), and their metabolites play an important role in the synthesis of various inflammatory mediators, such as prostaglandin (PG), leukotrienes (LT), thromboxanes (TX), protectins.”

10] Line 84: some references should be added on “cell membrane fluidity”.

RESPONSE: We have added the reference to “cell membrane fluidity”.

“Moreover, N3PUFA is essential in regulating lipid rafts and influencing cell membrane fluidity [16].”

11] Line 119: “…or over 50%...” improve syntax.

RESPONSE: We have improved syntax accordingly.

“While 81% of symptomatic individuals had the relatively mild disease after infection, 14% developed severe disease with dyspnea, hypoxia, or lung involvement over 50% by imaging; and, 5% developed critical disease with respiratory failure, shock, and/or multiorgan dysfunction [25].”

12] Line 172: put “oh” into the correct form.

RESPONSE: We have corrected to “on”.

“IL-6 also increases the expression of TF on monocytes and endothelial cells to further worsen under failure dysfunction [45].”

13] Line 196: correct the phrase “…program to sell deaths…”.

RESPONSE: We have corrected to “cell”.

“The viral infection can also induce vascular injury leading to pyroptosis, which is a program to cell deaths thatch release various damage-associated molecular patterns (DAMPs) and pathogen-associated molecular patterns (PAMPs) [47].”

14] Line 213: correct the phrase “…to reduce the improve…”.

RESPONSE: We have corrected the sentence.

“N3PUFA has been shown to improve damaged endothelial function through various mechanisms.”

15] The subtitles of sections 3.1, 3.2, and 3.3 should be the same as the entries of the pathological significance column of Table 1.

RESPONSE: We have changed the subtitles of sections 3.1, 3.2, and 3.3 accordingly to “N3PUFA and endothelial function”, “N3PUFA and immunothrombosis”, and “N3PUFA and inflammation”.

16] Line 236: how much was the high intake?

RESPONSE: We have added the high intake value to the sentense.

“The 2018 epidemiology study observed that a diet including a high intake of marine N3PUFA (≥ 4.7 g/week) was associated with a reduced risk of venous thromboembolism (VTE) following unprovoked index events [59].”

17] Line 238: correct to “meta-analyses”.

RESPONSE: We have corrected the sentense.

“Again, meta-analysis of prospective cohort studies confirmed N3PUFA’s protective effect on VTE and recurrent VTE [60].”

18] Line 290: change 3.2 to 3.3

RESPONSE: We have corrected the section number to 3.3.

19] Line 343: change “intestinal” to “intestine”. The acronym FFA should be explained in the first appearance.

RESPONSE: We have corrected the word and added explaionation of FFAs.

“Following ingestion, N3PUFA is hydrolyzed, like the other dietary lipids, in the intestine to FFAs (free fatty acids) and monoglycerides, which can then be incorporated into micelles after bile salt emulsification and absorbed into enterocytes by passive diffusion into chylomicrons [84].”

20] Line 353: change into a new paragraph.

RESPONSE: We changed into a new paragraph.

21] Line 361: improve syntax.

RESPONSE: We have improved the syntax.

“In an acute study, the absorption of EPA in TG form is 90% and 60% in EE form [85].”

22] Line 379: it seems that “anti-oxidative” should read “oxidative”.

RESPONSE: We have corrected the word.

“N3PUFA has conferred cardiovascular health by reducing inflammation, oxidative stress, improving arterial and endothelial functions, and reducing platelet aggregation [79].”

23] Line 397: improve syntax.

RESPONSE: We have corrected the syntax.

“Therefore, to establish the dose that can demonstrate clinically significant cardiovascular benefit, previous RCTs that studied the association between N3PUFA and CVD have been reviewed.”

24] Table 2: Leave a blank line between studies to improve the readability in the “Result” column.

RESPONSE: We have added a blank line between studies to improve the readability.

25] Line 403: HH should read HR.

RESPONSE: We have corrected HH to HR in table

26] Line 448: HA should read HR.

RESPONSE: We have corrected HA to HR.

“Although the VITAL study of the US cohort found that daily administration of 2000 IU/day of vitamin D3 and 1 g/day of N3PUFA (combination of EPA and DHA) did not significantly reduce the risk of CVD when compared to the placebo group after 5.3 years of intervention [108], a statistically significant decrease in HR of MI was re-ported.”

27] Line 460: “Table 1” should read “Table 2”.

RESPONSE: We have corrected Table 1 to Table 2.

“Moreover, the findings from later studies using the low dose N3PUFA (0.376 – 1 g/day) failed to demonstrate its sufficiency for populations who have lower average fatty fish consumption to reach the same therapeutic benefit to lower cardiovascular risk [100–105,107,108] (Table 2).”

28] Line 484: The “baseline N3PUFA index” should be defined. A citation is missing.

RESPONSE: We have defined “N3PUFA index” in the context. A citation has been added.

“A recent update on the dose recommendation of N3PUFA reviewed the threshold of baseline N3PUFA index (a calculated value to express the amount of systemic EPA and DHA in weight (%) present in the tissue cell membrane lipid fraction), and the supplementation dose has suggested that high-dose N3PUFA (4 g/day) appears to be more beneficial among the people with low baseline N3PUFA (< 8% N3PUFA index, a measurement of serum N3PUFA level) [113].”

29] Line 506: in the phrase “oil/N3PUFA PUFAs” the “PUFAs” is redundant.

RESPONSE: We have deleted PUFAs in the sentence.

“It is, therefore, prudent to study the possible uses of fish oil/N3PUFA supplementation as an adjuvant to medication in COVID-19 patients at risk for vascular thrombotic events.”

Round 2

Reviewer 2 Report

Now the manuscript is significantly improved.

There is still a concern about the Section 3.1 title which seems to be missing.

Author Response

Response to editors and reviewers:

We would like to thank the reviewer's second round comment. Please find below our response to the reviewers in blue. The changes have been made in the version 3 manuscript.

The subtitle of section 3.1 has been revised accordingly to "N3PUFA and endothelial function".